# The Use of Medical and Non-Medical Services by the Elderly during the SARS-CoV-2 Pandemic Differs between General and Specialist Practice: A One-Center Study in Poland

**DOI:** 10.3390/healthcare9010008

**Published:** 2020-12-23

**Authors:** Justyna Mazurek, Karolina Biernat, Natalia Kuciel, Katarzyna Hap, Edyta Sutkowska

**Affiliations:** Department and Division of Medical Rehabilitation, Wroclaw Medical University, 50-556 Wroclaw, Poland; justyna.mazurek@umed.wroc.pl (J.M.); natalia.kuciel@umed.wroc.pl (N.K.); edyta.sutkowska@umed.wroc.pl (E.S.)

**Keywords:** COVID-19, SARS-CoV-2, coronavirus pandemic, healthcare needs, assessment, geriatrics, quality of care, care coordination, decision-making, health professionals

## Abstract

In Poland, there is a lack of documented data on the use of medical and non-medical services by the elderly during the SARS-CoV-2 pandemic. The FIMA questionnaire assesses the use of medical and non-medical services by the elderly. The authors compared the demand for these services during the ongoing pandemic with similar months in 2017. It was confirmed that in the group of 61 surveyed elderly people, the number of individuals who had a medical visit decreased significantly in the three-month period. In the analyzed pandemic period, patients had significantly fewer visits to their general practitioner only. The pandemic had no significant impact on the use of other medical and non-medical services analyzed by FIMA. The limitations may include the small number of respondents, the relatively short period from the beginning of the pandemic covered by the survey, and the nature of the studied patients’ diseases. Further observation of elderly patients’ access to the abovementioned services can improve the efforts of governments and caregivers in this field, which is of particular importance in the group of chronically ill elderly patients.

## 1. Introduction

Coronavirus disease 2019 (COVID-19) is caused by a new virus, i.e., severe acute respiratory syndrome coronavirus 2 (SARS-CoV-2) [1,2]. Due to the much more aggressive nature of the infection in terms of its signs and symptoms [3,4,5], the specific decision was made by national governments to stop the spread of the virus. Social distancing was one of the protective steps, recommended as the key measure that could help to control the epidemic [2]. However, regarding the main places where people can become infected, not only restaurants or shops closed down, but medical clinics also reduced the number of their consultations or introduced phone consultations and videoconferencing instead [6]. People have also avoided contact with medical staff, except for emergencies [7,8,9,10].

COVID-19 is overwhelming healthcare services and healthcare professional workers globally. This situation mostly affected the elderly and patients with chronic diseases, as those are particularly vulnerable groups during the pandemic [11]. Despite the highest number of deaths in that group, in several countries so far, limitations concerning geriatric medicine (geriatrics) have been observed. They involved reduced access both to medical facilities and to physiotherapy services. “The issue with care coordination is that providers may not be able to manage both medical and nonmedical issues, such as making sure patients see all their doctors and specialists, and connecting them with community-based support organizations” [12]. Limited opportunities for teamwork, social contacts, and access to psychiatric care have also been defined as a problematic issue [13,14]. In France, for instance, the new situation shook up all medical practices; since the onset, all face-to-face hospital consultations were suspended, except immediately life-threatening pathology or uncontrolled pain [15]. It is also worth emphasizing that the lengths of stay in rehabilitation departments during the COVID-19 pandemic are much shorter in several countries, including Belgium, India, Tanzania, and the United Kingdom [16]. Data from the Office for National Statistics in the United Kingdom shows that 31% of people over 70 say that their access to healthcare and treatment for non-coronavirus related issues is being affected, and that they have been particularly affected by the reduction in face-to-face GP appointments [17]. Different national organizations are creating toolkits to address the elderly’s increased demand for information about where to find resources—most of all, healthcare and medication [18,19].

Moreover, during quarantine, elderly people who lived alone were more likely to have difficulties related to the supply of food and drugs, and experienced difficulties accessing home services (e.g., home care) and transport [20]. It was, and still seems to be, a new situation that can change the previously known medical and non-medical market, and can have a significant impact on the shape and development of medical and non-medical services [11].

In Poland, there is a lack of documented data on the use of medical and non-medical services by the elderly during the SARS-CoV-2 pandemic.

To investigate whether these new circumstances create a new model of services, we have assessed the use of medical and non-medical services by the elderly during the SARS-CoV-2 pandemic in Poland, and decided to compare it to that before the pandemic. We applied the FIMA (Fragebogen zur Inanspruchnahme medicinisher und nicht-medizinisher Versorgungsleistungen im Alter; Eng. Questionnaire for Health-Related Resource Use in an Elderly Population) questionnaire, which is the most suitable tool for such an assessment, as it is characterized by high accuracy and reliability [21]. The original FIMA questionnaire contains 28 questions. The first eighteen questions concern the medical and paramedical services used by the elderly, while the further questions (19–28) focus on the socio-demographic data and assess the degree of difficulty and time needed to complete the questionnaire by the patient (Table 1). Because of differences in the Polish model of health insurance, questions 17 and 18 are not included in the Polish version of FIMA. That version was successfully validated by our team three years ago [22].

Healthcare in Poland is insurance-based, and is delivered through a publicly funded healthcare system called the National Health Fund (NHF). Citizens are obligated to pay an insurance fee (redistributed tax; usually they have their health insurance paid for by their employer, or are the spouse or child of an insured person), which is 9% deducted from personal income (7.75% is deducted from the tax, and 1.25% covered by the insured goes directly to the National Health Fund). According to Article 68 of the Polish Constitution [23], everyone has a right to have access to healthcare. In particular, the government is obliged to provide free healthcare to the elderly, young children, pregnant women, and disabled people. Access to healthcare services can be provided if the patient is able to confirm having health insurance by presenting a document, such as an insurance card, an insurance card for employee family members, or a pensionary card.

## 2. Materials and Methods

### 2.1. Participants and Procedure

The participants were recruited in the period from July to August 2020, following the conclusion of the agreement (No. 411/2020) with the Bioethics Committee of the Wroclaw Medical University. All of the patients were informed about the aim of the study and their rights, and they gave oral permission for participation in the study.

Elderly people (age > 65) consecutively admitted to the rehabilitation ward at the St. Jadwiga Hospital in Trzebnica were asked to participate and fill in the FIMA questionnaire within 24 h after admission, until the number of participants was 61. This was the number derived from the limitation described below. The rehabilitation ward includes 84 beds, and is intended for the treatment and rehabilitation of people, mainly the elderly, (1) who have undergone surgery in different areas of the musculoskeletal system (e.g., after joint arthroplasty, fractures, or multi-organ injuries), or (2) with chronic diseases (including, in particular: discopathy, spondylosis, rheumatoid arthritis, or osteoarthritis). The stay in the ward lasts from four to six weeks.

The exclusion criteria included severe eye or ear diseases, a life-threatening condition, cognitive impairment preventing the patient from understanding all of the FIMA questions (assessed by the Mini-Mental State Examination—MMSE < 26 pts), and refusal to participate in the study.

The results from the FIMA questionnaires obtained during the SARS-CoV-2 pandemic (study group) were subsequently compared to the results from the corresponding group of patients who filled in the FIMA during their hospital stay between July and August 2017 (comparison group).

In 2017 and 2020, 72 and 69 elderly people from among those who consecutively reported to the rehabilitation department during the aforementioned period, respectively, were asked to fill in the FIMA questionnaire. The inclusion criterion for the study, that is MMSE ≥ 26 pts, was met by 68 and 65 people in 2017 and 2020, respectively. In 2017, seven people refused to fill in the FIMA questionnaire after reviewing its contents, despite their prior consent. In 2020, four patients withdrew their consent.

It should be noted that the recruitment of elderly people for the study during the SARS-CoV-2 pandemic lasted until reaching the same number of questionnaires as in 2017 (*n* = 61), so it was limited by the number of FIMA documents completed three years ago. Thus, the study covered the same number of participants (61 individuals) from the same rehabilitation department.

### 2.2. Statistical Analysis

Statistical analysis was performed using the R statistical package (R for Windows v. 4.0.2) [24]. Differences between the compared groups were assessed using the Mann–Whitney U test for continuous variables (number of doctor’s visits, age, etc.) and the chi-squared test of independence for categorical variables. The effect size was calculated using the Hedges’ g or odds ratios with 95% confidence intervals (CIs). The normality of the data was assessed using the Shapiro–Wilk test and graphical evaluation (histogram, QQ plot). For all statistical tests, *p* < 0.05 was considered significant.

## 3. Results

Sixty-one participants were recruited during the period concerned. The number of participants was limited to that included in the available archival data from 2017, which was used for comparison. The mean age was 71.4 (SD = ± 5.7) in the study group and 73.0 (SD = ± 6.0) in the comparison group. Women dominated in both groups, the majority of the elderly patients had secondary education, and more than half of them were in a steady relationship. The compared groups did not differ in terms of socio-demographic characteristics (Table 2).

In both groups, the majority of respondents rated the difficulty of the FIMA questionnaire as simple (study group: 60.71%, *n* = 37 and comparison group: 54.13%, *n* = 33). There was no statistically significant difference between the groups in terms of the time needed to complete the FIMA (study group: 9.00 min and comparison group: 7.30 min).

Generally, a statistically significant lower number of the patients visited doctors during the pandemic period (one part of the FIMA questionnaire; *p* < 0.001; OR: 0.09; CI: 0.01–0.33, comparison group as baseline), however, only the number of visits to the general practitioner was found to be reduced (*p* < 0.004; OR: 0.55; CI: 0.08–0.99, comparison group as baseline) during the COVID-19 pandemic compared to the corresponding period in 2017. Regarding other components of the FIMA, no significant differences were observed. It is notable that the mean number of drugs taken daily among the surveyed seniors amounted to be as many as 6.62 (SD = 3.41) in the study group and 6.20 (SD = 3.30) in the comparison group, with no statistically significant difference between them. The detailed results obtained from the FIMA questionnaire are presented in Table 3.

## 4. Discussion

The healthcare system in Poland and worldwide is going through one of the most difficult tests in the history of modern medicine. The impact of the COVID-19 pandemic on basic medical services worldwide is a source of deep concern for healthcare and related institutions. The progress of medicine and healthcare achieved over the last few decades can be quickly lost, as was already observed in the past, e.g., during armed conflicts or Ebola outbreaks [25,26]. However, as the WHO emphasizes, a new problem, which has not been noticed so far, will be the necessity to face the consequences of many omissions in the diagnosis and treatment of known diseases that have been driven to the background in the fear of coronavirus infection or postponed in the face of the imposed restrictions related to direct contact [27].

In August 2020, the WHO published a report on the survey conducted among health ministry officials from over 100 countries during the pandemic. A preliminary assessment of the impact of the COVID-19 pandemic on health services was carried out in the study. The vast majority of the surveyed countries reported disruptions in access to basic health services, and presented their own experience in looking for an appropriate strategy of service provision during the COVID-19 pandemic. Their analysis can help understand the source of disruptions and develop a global strategy that will enable the maintenance of basic health services at the necessary level throughout the pandemic [28]. This is the first study in Poland that presents the results of a comprehensive assessment of seniors’ access to medical services (outpatient and inpatient care) and non-medical services.

Our study aimed to answer the question whether there were any significant changes in the use of medical and non-medical services among the elderly during the period of restrictions introduced in the first few months of the SARS-CoV-2 pandemic in Poland. The results obtained from the FIMA questionnaire conducted during the pandemic were compared to the results obtained three years ago in the same center, i.e., the rehabilitation department in Trzebnica, for a similar group of patients.

The observed difference during the pandemic was that a fewer number of people had medical visits during this period in general. In particular, fewer GP visits were noted during three months of the pandemic (*p* < 0.004; CI = 0.547) compared to similar months three years before. These results reflect the limitation of personal contact between the patients and their GPs during the COVID-19 pandemic, which is imposed by the government. It should be emphasized that other medical services, both inpatient and outpatient, or the average number of visits to specialists except the GP, did not change.

Although the epidemic caused by the SARS-CoV-2 virus mobilized the healthcare system to widely use phone consultations, the main problem in the several initial weeks was the lack of possibility of a smooth transition from standard visits to consultations via phone. This was particularly related to the difficulty in patients’ adaptation to the new situation and the new type of consultations. As a result, some of the standard services performed by primary healthcare units before the pandemic were transferred to hospitals (more precisely, to accident and emergency units), or the visits were simply abandoned by the patients, which is probably reflected in the results of the survey. Perhaps the reduction in the number of visits, with regard to the population of elderly people, was partly due to technical difficulties encountered by this group of patients when using services via ICT. The new situation and the new rules associated with it, imposed overnight on a sort of compulsory basis due to epidemic risks, are also often difficult for elderly people to accept. Regarding the unquestionable usefulness of phone consultations or videoconferencing, it should be noted that this form of consultation can replace the traditional ones in the scope of such simple services as obtaining an electronic prescription or an electronic order for medical devices and online sick notes, but often cannot replace the physical examination of the patient. This applies especially to the so-called first-time visits and those where a medical examination is necessary. While in some medical specialties, a periodic interpretation of laboratory test only results seems sufficient (e.g., an assessment of the glycemic profile by an endocrinologist in a patient with diabetes), a phone call cannot replace e.g., an abdominal examination in a patient who reports abdominal pain. Thus, further evaluation is required to find out the scale of the described phenomenon of the abandonment of visits or their replacement with phone consultations, and to discover the health consequences of this, as those aspects have not been covered by our study.

To sum up, for some services, telemedicine may not always be a sufficient tool of dealing with patients, and it seems understandable that some patients use it in a limited scope. The respondents were patients of the rehabilitation department, so they were a selected group of elderly people. Despite a visible reduction in the number of visits to a GP, no statistically significant reduction in the number of physiotherapy services was noted. Given the group of patients subject to evaluation, this element surprised us. The observed fact may be explained by the underestimation resulting from a small, limited size of the ward (42 beds) and the number of respondents, which may have affected the *p*-value. Moreover, it should be taken into account that rehabilitation wards, where locomotor disorders are treated, are most frequently visited by patients with chronic complaints. These people often do, for example, get recommended exercise at home while awaiting hospitalization, or are consulted every few months to assess the effectiveness of the therapy or to correct recommendations. Therefore, the period of three months covered by the FIMA may not have been sufficient to note the differences. Patients could consult possible doubts with a physiotherapist during the proposed teleconsultations via ICT systems, which were introduced by the Ministry of Health in Poland (the FIMA does not differentiate between the type of consultation, i.e., standard or phone consultation). Physiotherapists conducted video consultations and online exercises with patients. Thus, the group surveyed might not have felt the limitation of personal contacts in their therapy as much as, for example, people with acute conditions who report to their GP in the first place. In addition, most people who are chronically ill are determined to treat their most afflictive condition (in this case, locomotor ailments) and find various ways to get consultation despite the restrictions imposed, e.g., if they cannot visit the clinic for a massage (paid or unpaid) due to its closure, they use the service at home (even if it is prohibited by the restrictions). The abovementioned possibilities of using physiotherapy services may explain the lack of significance between the groups in the analyzed period of three months. Additional facilitation in the scope of physiotherapeutic services introduced several years ago is a possibility for a physiotherapist to make independent therapeutic decisions about rehabilitation activities (without a referral from a doctor).

According to first publications, COVID-19 has had a profound impact on the organization of rehabilitation in other countries. In their article, Vigorito C. et al. stress the importance of the impact of the COVID-19 pandemic on the rehabilitation of cardiac patients in Italy. The authors reported a significantly reduced number of patients rehabilitated in specialist centers. This is due to the reduced number of referrals from specialist physicians and the postponement of many planned interventions on patients in coronary care and intensive care units. The total number of patient admissions has been significantly reduced, also due to the abandonment of scheduled treatments by patients themselves [29]. The first projects are currently underway to restore the continuity of rehabilitation services despite the pandemic. In their article, Brettger et al. described adjustments in the scope of continuity of rehabilitation services introduced in 12 countries during the ongoing pandemic in response to the initial suspension or significant decrease in efficiency of these services [16]. The proposed solutions include tele-rehabilitation, which aims to provide continuous counseling, education, and development of an individual e-mail exercise program for patients. Teleconsultations take place through phone and video, and are frequently conducted using free applications (e.g., WhatsApp, Zoom, WeChat). Standards of rehabilitation care are constantly changing. The abovementioned uses are aimed at alleviating the effects of the pandemic on physiotherapeutic care and at supporting patients’ health with the lowest risk of disability.

Regarding the general consequences of restricting access to medical services, which have been observed in the smaller number of people who benefited from consultations in the first few months of the pandemic, the reports on the impact of the epidemic regime on access to health professionals in Italy should be taken into account [13]. The primary goal in that country was to isolate COVID-19 patients and provide them with appropriate care. This re-evaluation of medical activities particularly affected the geriatric population not suffering from COVID-19, as their access to most healthcare system resources was significantly reduced. This was due to the lack of a sufficient number of medical personnel and the need to protect the elderly, who are most vulnerable to COVID-19′s fatal complications. One of the first assessments of the impact of the pandemic on the lives of chronically ill patients showed that the limited access to diagnostic tests due to the ongoing pandemic will result in a significant number of additional deaths caused by breast, esophageal, lung, and colorectal cancers [30]. The Polish Society of Oncology also alerts that the effects of “lockdown” in our country will undoubtedly delay the diagnosis of cancer in many patients. A delayed diagnosis of the disease will result in a higher severity of the disease at the time of diagnosis and, as a consequence, its more difficult, more expensive, and less effective treatment.

In summary, our assessment of changes in the use of medical and non-medical services by seniors, expressed by means of the FIMA questionnaire, confirmed the significant reduction in visits of the elderly to a GP. Despite the profile of diseases of hospitalized patients, no significantly reduced number of physiotherapeutic services was noted with regard to our respondents before hospitalization, despite the lockdown. The explanation of this fact can only be hypothetical, since the FIMA does not investigate the causes. In this respect, it seems to us that, in the case of specific chronic conditions, patients are both more determined to seek help despite the lockdown regime and have more skills to cope with this difficult situation. We cannot rule out either that this group of seniors succeeded in the use of the new form of therapy through teleconsultations, although this seems unlikely. Although the analyzed group represented a single disease profile (taking into account diseases related to the locomotor system), by analyzing their age, we can assume that this population shows a typical cross-section of the geriatric society of the Polish population.

Bearing in mind the consequences of the lack of regular treatment of chronic diseases, it is necessary to continue to monitor the situation on the market in terms of the demand for the abovementioned medical and non-medical services. This will not only allow examining patients’ compliance in the long term, when avoiding the coronavirus infection overshadows other health goals, but may also contribute to better organization of patients’ future access to these services.

## 5. Conclusions

1. Fewer visits to the general practitioner were recorded.

2. The 3–4 month period of social distancing regime did not significantly affect the use of other medical and non-medical services among the studied group of seniors.

## 6. Limitations

Considering the aging population in Poland, the study group is relatively small. The study was limited to the patients of one large centre, which may not fully reflect the range of medical and non-medical use and the scope of support in other regions or cities in Poland, particularly in the country. The short time of the assessed period may not be representative of long-term limitations in access to medical and non-medical services. In addition, response bias, which is the unconscious influence of the respondent on the anamnesis and the obtained results, cannot be excluded. Another limitation of the presented study is the possible sample bias of people who are willing to visit a provider during a pandemic, versus those that were willing to visit a provider in 2017.

## Figures and Tables

**Table 1 healthcare-09-00008-t001:** FIMA: cost categories, variables, units, and time periods [22].

Cost Category	Variable	Unit	Period
ambulatory health care	GP, internist, internist with specialty (e.g., cardiologist, gastroenterologist, nephrologist, diabetologist), gynaecologist, surgeon, orthopaedist, neurologist, dermatologist, ophthalmologist, urologist, dentist, psychotherapist, ambulatory stays at hospital, others	number of appointments (including the collection * of prescriptions and home visits)	Three months
remedies	physiotherapy (kinesiotherapy, massage, physical therapy), ergotherapy, medical pedicure, osteopathy	number of appointments	Three months
nursing and home care services	ambulatory community care, private nursing care, informal care (family, friends, etc.), short-term care	number of days in a week/number of days in a month/number of hours and minutes a day	Three months
medicines	name of drug, packaging size	daily/weekly/monthly/yearly dosage	Seven days
rehabilitation	ambulatory and stationary stays	number of days	12 months
stays in hospital (ambulatory and stationary)	outpatient clinics, stationary stays (including stays at the psychiatric hospital)	number of days	12 months
auxiliary equipment	walking frame, glasses, hearing device, dental prosthesis, breathing apparatus, wheelchair, hygiene pads, bath seat, others	ownership and usage	currently and within the last 12 months
relocations	necessity to change the place of residence	yes or no	12 months
type of residence	private household, shelter flat, residential care, others	yes or no	12 months

Abbreviations: FIMA—Questionnaire for Health-Related Resource Use in an Elderly Population; GP—general practitioner. * “Collection of prescriptions” means that the mean number of outpatient visits assessed using the FIMA questionnaire also includes those where the patient comes only to get a prescription. It does not apply to other questions.

**Table 2 healthcare-09-00008-t002:** Baseline characteristics for both groups.

Characteristic	Comparison Group (2017)	Study Group (2020)	*p*-Value	Effect Size (95% CI)
	N = 61	N = 61		
Mean age (SD)	73.00 (6.00)	71.40 (5.70)	0.16	0.26 (−0.10–0.64)
Sex n (%)			0.27	
Female	45 (74.00)	51 (84.00)		-
Male	16 (26.00)	10 (16.00)		0.55 (0.22–1.32)
Marital status n (%)			0.46	
In relationship	34 (56.00)	39 (64.00)		-
Single	27 (44.00)	22 (36.00)		0.71 (0.34–1.47)
Education n (%)			0.44	
Primary	6 (9.80)	8 (13.00)		-
Secondary	33 (54.00)	29 (48.00)		0.66 (0.20–2.21)
Vocational	11 (18.00)	17 (28.00)		1.16 (0.01–1.15)
Higher education	11 (18)	7 (11)		0.48 (0.11–1.95)

Abbreviations: SD—standard deviation; *n*—number of individuals in %; CI—confidence intervals.

**Table 3 healthcare-09-00008-t003:** Detailed FIMA questionnaire results for both groups.

Characteristics	Comparison Group (2017)	Study Group (2020)	*p*-Value	Effect Size (95% CI)
	N = 61	N = 61		
FIMA 1; n (%) ambulatory appointments with a doctor (three months)	59.00 (97.00)	44.00 (72.00)	**<0.001**	**0.09** **(0.01–0.33)**
GP; mean (SD)	2.44 (1.26)	1.77 (1.17)	**0.004**	**0.55** **(0.08–0.99)**
internist; mean (SD)	1.94 (1.26)	1.33 (0.65)		
gynecologist; mean (SD)	1.00 (0.00)	1.00 (0.00)		
surgeon; mean (SD)	1.50 (0.71)	1.33 (0.58)		
orthopedist; mean (SD)	1.42 (0.71)	1.15 (0.38)		
neurologist; mean (SD)	1.31 (0.48)	1.00 (0.00)		
dermatologist; mean (SD)	1.60 (0.89)	1.33 (0.58)		
ophthalmologist; mean (SD)	1.47 (1.17)	1.20 (0.45)		NSD
urologist; mean (SD)	1.00 (-)	1.00 (-)		
dentist; mean (SD)	1.90 (1.52)	1.86 (1.46)		
psychologist; mean (SD)	1.67 (1.15)	2.00 (-)		
stay at accident and emergency department; mean (SD)	1.25 (0.46)	1.60 (0.55)		
any other specialists; mean (SD)	1.00 (-)	1.00 (-)		
FIMA 2; n (%) ambulatory appointments with therapists (three months)	18.00 (30.00)	12.00 (20.00)		
physiotherapist mean (SD)	8.00 (6.00)	7.00 (11.00)		NSD for FIMA 2–17
ergotherapist; mean (SD)	1.00 (-)	0.00 (-)		
speech therapist; mean (SD)	0.00 (-)	0.00 (-)		
medical pedicure; mean (SD)	1.00 (-)	2.00 (-)		
FIMA 3; n (%) help of a community nurse/social worker (three months, average number of visits per month)	3.00 (4.90)	6.00 (9.80)		
visits per month; mean (SD)	3.00 (3.46)	4.17 (3.82)		
FIMA 4; n (%) use of paid care services (three months, average number of visits per month)	1.00 (1.61)	2.00 (3.32)		
visits per month; mean (SD)	30.00 (-)	1.00 (-)		
FIMA 5; n(%) use of informal help (three months, average number of visits per month)	23.00 (38.00)	27.00 (45.00)		
visits per month; mean (SD)	21.00 (11.00)	20.00 (11.00)		
FIMA 6; n (%) use of day residential care (three months, average number of days)	0.00 (-)	0.00 (-)		
number of days; mean (SD)	0.00 (-)	0.00 (-)		
FIMA 7; n (%) use of short-term care (three months, average number of days)	2.00 (3.30)	0.00 (-)		
number of days; mean (SD)	21.52 (12.00)	0.00 (-)		
FIMA 8; n (%) use of care benefits (type of benefits), in Polish zlotys (PLN) per month	10.00 (16.00)	5.00 (8.20)		
in polish zlotys; mean (SD)	295.00 (217.00)	599.00 (953.00)		
FIMA 9; n (%) medications taken	57.00 (93.00)	58.00 (95.00)		
number of medications; mean (SD)	6.20 (3.30)	6.62 (3.41)		
FIMA 10; n (%) use of stationary rehabilitation (12 months, average number of days/year)	28.00 (46.00)	22.00 (36.00)		
number of days per year; mean (SD)	1.57 (0.69)	1.74 (0.56)		
FIMA 11; n (%) undergone ambulatory operations (12 months, average number of operations)	6.00 (9.80)	5.00 (8.24)		
number of medications; mean (SD)	1.50 (0.84)	3.00 (1.41)		
FIMA 12; n (%) stationary stays at hospital wards (12 months, average number of stays/days)	29.00 (48.00)	27.00 (44.00)		
number of stays; mean (SD)	1.45 (0.74)	1.48 (0.80)		
number of days per year; mean (SD)	11.00 (6.00)	13.00 (11.00)		
FIMA 13; n (%) stationary stays at the psychiatric ward (12 months, average number of stays/days)	0.00 (-)	0.00 (-)		
number of stays; mean (SD)	0.00 (-)	0.00 (-)		
FIMA 14; n (%) auxiliary equipment	58.00 (95.00)	58.00 (95.00)		
rotator; mean (SD)	1.80 (0.42)	1.67 (0.50)		
wheelchair; mean (SD)	2.00 (-)	1.67 (0.58)		
crutches; mean (SD)	1.50 (0.51)	1.67 (0.48)		
bath seat; mean (SD)	1.20 (0.45)	1.50 (0.55)		
glasses; mean (SD)	1.05 (0.22)	1.02 (0.15)		
hearing device; mean (SD)	1.00 (0.00)	1.00 (-)		
dental prothesis; mean (SD)	1.09 (0.29)	1.02 (0.16)		
breathing apparatus; mean (SD)	0.00 (-)	1 (-)		
compression stockings; mean (SD)	1.33 (0.58)	1.11 (0.33)		
hygiene pads; mean (SD)	1.00 (-)	1.00 (-)		
Others; mean (SD)	1.33 (0.52)	1.67 (0.58)		
FIMA 15; n (%) type of residence (12 months) private household;	61.00 (100.00)	61.00 (100.00)		
FIMA 16; n (%) relocations (12 months) yes	1.00 (1.60)	1.00 (1.60)		
FIMA 17; n (%) health insurance yes	61.00 (100.00)	61.00 (100.00)		

Abbreviations: FIMA—Questionnaire for Health-Related Resource Use in the Elderly Population; SD—standard deviation; *n*—number of individuals in %; CI—confidence interval; NSD—no statistically significant difference.

## Data Availability

MDPI Research Data Policies.

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
