# Peer review of "The Use of Medical and Non-Medical Services by the Elderly during the SARS-CoV-2 Pandemic Differs between General and Specialist Practice: A One-Center Study in Poland"

_healthcare, 2020, doi:10.3390/healthcare9010008_

Round 1

Reviewer 1 Report

Thank you for the opportunity to review this article. The introduction and discussion are well-written, but the methods and results need much more description. You might be able to simplify by focusing on only a few of the FIMA questions, and mention that you did not find statistically significant differences between the two samples for any of the others. The small sample size is a major limitation of this work. 

L51 You state that “in Poland, there is a lack of scientifically documented data on the use of medical and non-medical services by the elderly” during the pandemic. It might be helpful to include any citations and a brief description of what you’ve found regarding elderly utilization elsewhere in the world in a few citations.

L63 Briefly describe the Polish model of health insurance.

L64 Do you have a citation for the validation study?

L71 Include a better description of the ward, how large it is, where it is located, who its usual clients are.

L80 The group interviewed in 2017 is perhaps better referred to as a comparison group, rather than a control group.

L81-90 It doesn’t seem critical that you ended up with 61 complete responses in each year, but there is significant focus on this fact.

I would relocate some of the tables to be placed just after the paragraph within which they are mentioned.

For table 1, explain the impact of the “collection of prescriptions” and if it applies to other questions as well.

L109-110 Please report the CI as a range, not a singular number, or call them the effect size and include the CI.

The statistical methods producing the results for table 3 need to be explained better for each of the different questions, if you decide to keep so many of the FIMA questions in the table. It might be cleaner to simply include a list of other questions asked and state that no statistically significant differences were found for most of the less-interesting questions.

When using effect sizes, state the directionality of the effect more explicitly in your results section.

The results section needs much greater elaboration.

Also in table 3, stay consistent with your choice of number of decimals. For example, in FIMA 2 you have 1.00 mean medical pedicures in 2017, but 8 mean physiotherapist visits. Table 3 requires significant formatting for consistency.

L144-146 You make the claim that sociodemographic characteristics did not change in a statistically significant way between 2017 and 2020, thus, you claim the results can be attributed to pandemic conditions. This is not 100% accurate. Furthermore, in most cases it is likely that you have not found statistical significance due to your small sample size.

I am not a proponent of writing conclusions and limitations in bullet form.

Another important limitation that you did not list is the possible sample bias of people who are willing to visit a provider during a pandemic, versus those that were willing to visit a provider in 2017.

Author Response

REPLY TO COMMENTS OF THE REVIEWER 1

Reviewer #1:

We are grateful for the positive and constructive comments that originated in the review process. We have carefully reviewed the comments and have revised the manuscript accordingly. A major revision of the paper has been carried out to take all of them into account, and in the process, we believe the paper has been significantly improved.

 Our  responses are given in a point-by-point manner below. Changes to the manuscript are marked red.

Thank you for the opportunity to review this article. The introduction and discussion are well-written, but the methods and results need much more description. You might be able to simplify by focusing on only a few of the FIMA questions, and mention that you did not find statistically significant differences between the two samples for any of the others. The small sample size is a major limitation of this work.

RESPONSE: Thank You very much for Your revision. We have changed and described the methods and results parts in details.

L51 You state that “in Poland, there is a lack of scientifically documented data on the use of medical and non-medical services by the elderly” during the pandemic. It might be helpful to include any citations and a brief description of what you’ve found regarding elderly utilization elsewhere in the world in a few citations.

RESPONSE: Thank You for pointing this out. We have looked up in the literature some more citations and have described what we are found regarding elderly utilization elsewhere in the world.

L63 Briefly describe the Polish model of health insurance.

RESPONSE: Thank You for this remark. We have described the Polish model of health insurance in the Introduction part:

Health care in Poland is Insurance based and is delivered through a publicly funded health care system called the National Health Fund (NHF). Citizens are obligated to pay insurance fee (redistributed tax, usually they have their health insurance paid for by their employer, or are the spouse or child of an insured person) which is 9% deducted from personal income (7,75% is deducted from the tax, 1,25% covered by insured goes directly to the National Health Fund). According to Article 68 of the Polish Constitution everyone has a right to have access to health care. In particular, the government is obliged to provide free health care to the elderly, young children, pregnant women and disabled people. Access to health care services can be provided if patient is able to confirm having health insurance by presenting a document such as an Insurance card, an Insurance card for employee family members or a Pensionary card.

L64 Do you have a citation for the validation study?

RESPONSE: Yes, of course we have added a citation for the validation study. Sorry for this mistake:

Mazurek, J.; Sutkowska, E.; Szcześniak, D.; Urbańska, K.M.; Rymaszewska, J. FIMA, the questionnaire for health-related resource use in the elderly population: validity, reliability, and usage of the Polish version in clinical practice. Clin Interv Aging 2018, 13, 787-795.

L71 Include a better description of the ward, how large it is, where it is located, who its usual clients are.

RESPONSE: Thank You for this suggestion. We have included a better description of the rehabilitation ward:

The rehabilitation ward includes 84 beds and is intended for the treatment and rehabilitation of people, mainly elderly, 1/ who have undergone operations in different area of ​​the musculoskeletal system (eg. After joints arthroplasty, after fractures or multi-organ injuries), or 2/ with chronic diseases (including in particular: discopathy, spondylosis, rheumatoid arthritis, osteoarthritis). The stay in the ward lasts from 4 to 6 weeks.

L80 The group interviewed in 2017 is perhaps better referred to as a comparison group, rather than a control group.

RESPONSE: We agree with this remark. We have changed the nomenclature throughout the article. 

L81-90 It doesn’t seem critical that you ended up with 61 complete responses in each year, but there is significant focus on this fact.

RESPONSE: Yes, we agree, it doesn’t seem critical that we ended up with 61 complete responses, but we decided to include this information in the text, so that the potential reader has full insight into the patient recruitment process and the course of the study.

I would relocate some of the tables to be placed just after the paragraph within which they are mentioned.

RESPONSE: We also agree with this suggestion of the Reviewer and have relocated our tables accordingly.

For table 1, explain the impact of the “collection of prescriptions” and if it applies to other questions as well.

RESPONSE: Thank You for this important question. „Collection of prescriptions” means that the mean number of outpatient visits assessed using the FIMA questionnaire also includes those where the patient comes only to get a prescription. It dose not applies to other questions. We have explained it under the table 1.

L109-110 Please report the CI as a range, not a singular number, or call them the effect size and include the CI.

RESPONSE: Yes, we agree, we have reported the CI as a range.

The statistical methods producing the results for table 3 need to be explained better for each of the different questions, if you decide to keep so many of the FIMA questions in the table. It might be cleaner to simply include a list of other questions asked and state that no statistically significant differences were found for most of the less-interesting questions.

RESPONSE: Thank You very much for this important remark, we totally agree with it. To present our results in a clearer way we decided – if You agree - to leave a list of all FIMA questions in table 3 with no statistically significant differences, as the Reviewer suggests (we have removed all less interesting results). We think that it could be interesting for the potential reader to know the detailed FIMA questionnaire results for both groups: comparison and study group, so we left the result of the first two columns of table 3.

When using effect sizes, state the directionality of the effect more explicitly in your results section.

RESPONSE: We stated it more explicitly in our results section. Thank You.

The results section needs much greater elaboration.

RESPONSE: We have described our results more comprehensively, as the Reviewer suggests. Thank You. 

Also in table 3, stay consistent with your choice of number of decimals. For example, in FIMA 2 you have 1.00 mean medical pedicures in 2017, but 8 mean physiotherapist visits. Table 3 requires significant formatting for consistency.

RESPONSE: Yes, the Reviewer is right, we have corrected it. We hope, now our table 3 is more consistent and readable. Thank You.

L144-146 You make the claim that sociodemographic characteristics did not change in a statistically significant way between 2017 and 2020, thus, you claim the results can be attributed to pandemic conditions. This is not 100% accurate. Furthermore, in most cases it is likely that you have not found statistical significance due to your small sample size.

RESPONSE: We also agree with this important remark, so we decided to cancel this sentence.

I am not a proponent of writing conclusions and limitations in bullet form.

RESPONSE: We have changed the limitations part, so now it is no in bullet form anymore:

Considering the aging population in Poland, the study group is relatively small. The study was limited to the patients of one large centre, which may not fully reflect the range of medical and non-medical use and the scope of support in other regions or cities in Poland, particularly in the country. Short time of the assessed period may not be representative of long-term limitations in access to medical and non-medical services. In addition, response bias, which is the unconscious influence of the respondent on the anamnesis and the obtained results, cannot be excluded. Another limitation of the presented study is the possible sample bias of people who are willing to visit a provider during a pandemic, versus those that were willing to visit a provider in 2017.

Another important limitation that you did not list is the possible sample bias of people who are willing to visit a provider during a pandemic, versus those that were willing to visit a provider in 2017.

RESPONSE: This is correct, and we have included this important limitation of our study

Reviewer 2 Report

Line 33-34 as well as a much more likely life-threatening nature of the disease,

The mortality rate is lower. Delete.

“The data from this study supports the fact that the CFR [mortality rate] of the COVID-19 pandemic seems to be less than Bird flu, Ebola, SARS, and MERS…” (Khafaie & Rahim, 2020; p. 78)

Khafaie, M. A., & Rahim, F. (2020). Cross-country comparison of case fatality rates of COVID-19/SARS-COV-2. Osong Public Health and Research Perspectives11(2), 74.

Line 38 phone consultations also include videoconferencing (see Line 176 when you mention telemedicine and then Line 213-216)

Line 39-41  “The people's ability to perceive what is more harmful to them seems to be based on the information from media (television or Internet) that are closer to them than a general practitioner (GP) in a pandemic circumstances.”

Medical services restricted non-essential visits and operations. This decision to restrict access to primary care was directed from top down and patients did not have a say to access. You later introduced this directive in line 219.

Line 48, Include that older adults also experienced difficulties accessing home services (e.g., home care) and transportation.

Line 51 delete “scientifically” documented data is valid on its own (delete “scientifically”  also in the Abstract. There is no such thing as scientifically documented data, just the method of obtaining the information)

Line 190-191 It is a shame that “FIMA does not differentiate between the type of consultation, i.e. standard or phone consultation” this could have been useful to explore how the health care services (other than GP) adapted to the new pandemic conditions.

Line 227-232 edit to read :

One of the first assessments of the impact of the pandemic on the lives of chronically ill patients was made by the authors of a publication in "Lancet Oncology". They attempted to estimated that  the consequences of delayed oncological diagnosis in the UK. According to the authors of the study, the limited access to diagnostic tests due to the ongoing pandemic will result in a significant number of additional deaths caused by breast, oesophagal, lung and colorectal cancers [24].

Delete 254-255 1. “During the 3-4 months of "lockdown", a smaller number of the surveyed elderly people availed of medical services.”

Your results are the opposite as you point out in the following statement in number 3.

This is interesting primarily because of the lack of significant differences found. it could inform us of the role GPs have in the lives of older adults.

Author Response

REPLY TO COMMENTS OF THE REVIEWER 2

Reviewer #2:

We are grateful for the positive and constructive comments that originated in the review process. We have carefully reviewed the comments and have revised the manuscript accordingly. A revision of the paper has been carried out to take all of them into account, and in the process, we believe the paper has been significantly improved.

 Our responses are given in a point-by-point manner below. Changes to the manuscript are marked red.

Line 33-34 as well as a much more likely life-threatening nature of the disease,

The mortality rate is lower. Delete.

“The data from this study supports the fact that the CFR [mortality rate] of the COVID-19 pandemic seems to be less than Bird flu, Ebola, SARS, and MERS…” (Khafaie & Rahim, 2020; p. 78)

Khafaie, M. A., & Rahim, F. (2020). Cross-country comparison of case fatality rates of COVID-19/SARS-COV-2. Osong Public Health and Research Perspectives11(2), 74.

 RESPONSE: Thank You for this remark, of course we have deleted it.

Line 38 phone consultations also include videoconferencing (see Line 176 when you mention telemedicine and then Line 213-216)

 RESPONSE: Thank You for this suggestion, we agree with it and we have included also „videoconferencing” in the text.

Line 39-41 “The people's ability to perceive what is more harmful to them seems to be based on the information from media (television or Internet) that are closer to them than a general practitioner (GP) in a “pandemic circumstance.”

Medical services restricted non-essential visits and operations. This decision to restrict access to primary care was directed from top down and patients did not have a say to access. You later introduced this directive in line 219.

 RESPONSE: We agree with the Reviewer and we decided to delete this sentence.

Line 48, Include that older adults also experienced difficulties accessing home services (e.g., home care) and transportation.

 RESPONSE: Thank You, we included this information in the Introduction part.

Line 51 delete “scientifically” documented data is valid on its own (delete “scientifically” also in the Abstract. There is no such thing as scientifically documented data, just the method of obtaining the information)

 RESPONSE: Of course, we agree. We have deleted the word: scientifically.

Line 190-191 It is a shame that “FIMA does not differentiate between the type of consultation, i.e. standard or phone consultation” this could have been useful to explore how the health care services (other than GP) adapted to the new pandemic conditions.

 RESPONSE: Yes, this is a very important remark. Unfortunately the authors of the article do not have any influence of the shape of the FIMA questionnaire. But we have sent this interesting suggestion to the FIMA authors.

Line 227-232 edit to read :

One of the first assessments of the impact of the pandemic on the lives of chronically ill patients was made by the authors of a publication in "Lancet Oncology". They attempted to estimated that  the consequences of delayed oncological diagnosis in the UK. According to the authors of the study, the limited access to diagnostic tests due to the ongoing pandemic will result in a significant number of additional deaths caused by breast, oesophagal, lung and colorectal cancers [24].

 RESPONSE: We have edited this sentence. Thank You.

Delete 254-255 1. “During the 3-4 months of "lockdown", a smaller number of the surveyed elderly people availed of medical services.”

Your results are the opposite as you point out in the following statement in number 3.

RESPONSE: We have deleted the first conclusion, according to the Reviewers’ suggestion.

This is interesting primarily because of the lack of significant differences found. it could inform us of the role GPs have in the lives of older adults.

RESPONSE: Thank You.